# Does managerial myopia promote enterprises over-financialization? Evidence from listed firms in China

Yanchi Chen[1], Ju Ye[1]*, Qi Shi[2]

**1** Institute of Food and Strategic Reserves, Nanjing University of Finance and Economics, Nanjing, Jiangsu, China, **2** Economic School, Nanjing University of Finance and Economics, Nanjing, Jiangsu, China

* c774402944@126.com

**Data Availability Statement:** The data that support the findings of this study are available in http://doi.org/10.57760/sciencedb.10232, reference number https://www.scidb.cn/en/s/eEFzeq.

## Abstract

This paper analyzes the potential shortsightedness of enterprise managers through annual reports. Additionally, we use corporate financial statement data to measure enterprises over-financialization in terms of resource allocation. After testing with a causal inference model, we find that firms with managerial myopia significantly contribute to over-financialization. It remains robust even after the instrumental variable of whether the manager has experienced a famine is used. Furthermore, financial distress and financing constraints amplify the inclination of short-term-focused managers to amass greater financial assets.

## Introduction

Economic financialization has emerged as a significant driver behind the stagnation of economic growth and the decline in productivity [1]. This phenomenon is predominantly attributed to the escalating adoption of financialized practices among corporate entities, resulting in a squeeze on genuine investments and hindering their long-term growth trajectories [1, 2]. Moreover, the predominance of corporate financialization has impeded the overall development of the macroeconomy, exacerbating economic operational risks and restructuring of industrial sectors. Consequently, scholarly attention has been increasingly focused on elucidating the motivations driving corporate financialization to forestall the broader economic implications [3, 4]. However, on the contrary, corporate financialization is also propelled by motives of "reserving" and "profit-seeking", primarily aimed at mitigating financing constraints and engaging in speculative arbitrage. This not only aids in reducing the risks of funding interruption but also facilitates short-term high returns [5, 6], thereby favoring corporate development. Consequently, this study advocates for heightened scrutiny of corporate over-financialization, with managerial myopia identified as one of the primary catalysts.

It is crucial to clarify that the financialization behavior of corporations is closely associated with their unique financial conditions and operational performance, with managerial myopia being a key factor in causing fund squeezing and hindering long-term growth. Specifically, corporate financialization is a multifaceted concept, the economic outcomes of which require nuanced investigation. Taking high- and low-performance companies that hold financial

**Funding:** This work was supported by the [National Social Science Fund of China] under Grant [22VRC007]; [Institute of Food and Strategic Reserves, Nanjing University of Finance and Economics] under Grant [ BSZX2023-07]. The funders had no role in study design, data collection and analysis, decision to publish, or preparation of the manuscript.

**Competing interests:** The authors have declared that no competing interests exist.

assets as an example, high-performing enterprises adeptly address their investment needs, resulting in surplus idle funds, whereas low-performing enterprises exhibit more investment substitution. However, when corporations are led by myopic managers, they tend to prefer short-term, rapid investment projects [7], making short-term profit decisions at the expense of long-term interests [8], which evidently exacerbates the squeezing-oriented profit-seeking behavior of corporations. Nonetheless, existing research has tended to overemphasize the negative impacts of corporate financialization by exploring the motives and economic effects of financialization behavior from a homogenous perspective, thus neglecting its potential benefits [9, 10].

This paper wants to emphasize that enterprise financialization behavior is not equivalent to managerial myopia. Financialization is only a "double-edged sword" investment decision-making behavior motivated by resource allocation. Therefore, the financialization behavior of enterprises may not necessarily harm the long-term sustainable development of enterprises. Thus, this paper takes Chinese non-financial industry listed companies from 2005 to 2022 as samples, quantifying managerial myopia through the analysis of annual report texts; matches with the China Stock Market and Accounting Research Database (CSMAR) to obtain financial information for each company and identify indicators of corporate excessive financialization. Finally, using a mixed-effects OLS model and logistic model, it explores the impact of managerial myopia on corporate excessive financialization from the perspective of corporate governance. Empirical results indicate that Chinese non-financial listed companies generally engage in financial investment behavior. It is noteworthy that a considerable number of companies are found to exhibit a tendency towards excessive financialization, with the impact of managerial myopia becoming a key factor contributing to this phenomenon. Even after addressing potential endogeneity issues, we find the results to be robust. Additionally, the study also finds that the presence of internal financing constraints and financial distress exacerbates the tendency of myopic managers to engage in higher financial investment to smooth short-term benefits, while increasing the risk of excessive financialization.

The marginal contributions of this study are as follows: Firstly, we are among the first to construct an indicator of excessive financialization based on the perspective of corporate financial heterogeneity. This enriches research on corporate financialization, helping scholars to correctly grasp the dual nature of financialization and avoid its negative effects. Secondly, we consider the differences between managerial myopia and corporate financialization, empirically testing the phenomenon of excessive financialization caused by managerial myopia, helping shareholders better understand the underlying motives of financialization and achieve a balance between short-term and long-term interests. Finally, the study also considers, through moderation effect models, the impact of managerial myopia on corporate excessive financialization under external conditions such as financing constraints and financial distress, providing a theoretical basis for better corporate development.

The subsequent sections of this paper are organized as follows. Firstly, the "Literature Review" section provides a comprehensive overview of existing research on this topic. Secondly, the "Research Methodology" section delineates the model specifications and outlines the sample selection process applied in the empirical analysis. Following this, the "Empirical Analysis" section presents the findings obtained from the analysis and rigorously examines their robustness. Lastly, the "Conclusion" section summarizes and concludes the paper.

## Literature review

The relevant literature on the impact of managerial myopia on excessive corporate financialization mainly focuses on the effects of corporate financialization and its underlying logic, as

well as the measurement of managerial myopia and influencing factors. Concerns about the impact of corporate financialization are mainly concentrated in the macroeconomic domain among scholars. Epstein [11] defines financialization as the escalating influence of financial motives, markets, actors, and institutions on both domestic and international economies. Krippner [12], in contrast, characterizes financialization as a mode of accumulation wherein profits predominantly accrue through financial channels, rather than through conventional trade and commodity production. Unquestionably, the past four decades have witnessed a swift, substantial surge in financialization in the United States and globally, as defined by the above conceptualizations. This escalating trade of financialization has not gone unnoticed by researchers, who have unveiled its detrimental impact on overall economic growth. The proliferation of debt-based financial networks has compounded existing economic and social disparities [4, 13]. Similarly, Hein [14] contends that financialization is a primary driver of modern capitalist stagnation, intensifying the escalation of global macroeconomic risks. The phenomenon of macro-financialization embodies a concentrated expression of the pervasive micro-financialization, prompting scholars to delve into the behavioral rationale for corporate financialization and to reveal that the financialization of nonfinancial firms curtails tangible investment, thereby acting as a primary catalyst for sluggish economic growth and reduced productivity [1].

Subsequently, scholars have delved into analyzing the underlying behavioral logic of corporate financialization to achieve a reduction in corporate financialization [15, 16]. Entity firms engage in financial assets investment owing to a reserve motive or a profit-seeking motive [5, 6]. The reserve motive, rooted in Keynes' perspective [17], posits that maintaining liquid, realizable assets aids in alleviating funding constraints. Conversely, the profit-seeking motive arises from the emergence of financial markets, enabling firms to capitalize on carry and arbitrage trading opportunities driven by domestic currency appreciation [18]. Orhangazi [2] underscores that a manager may squeeze out the amount of their own fixed-asset investments owing the lure of financialized arbitrage, compromising long-term interests for short-term gains. In China, the real estate industry, which has a high degree of financialization, also faces the negative impact of debt risk shifting to banks, which is worthy of social and government vigilance [19]. Davis [20] presents opposing evidence, suggesting that financialization can stimulate fixed investment. Furthermore, corporate finance exhibits externalities, promoting innovation levels by alleviating financing constraints for other enterprises [21]. Therefore, societal concerns regarding corporate financialization primarily stem from its "profit-seeking" effects, rather than the "reserve" motives. Some scholars persist in using the financial asset ratio as an indicator of financialization, which may neglect the potential benefits of financialization and overly emphasize its drawbacks [10]. Effective resource planning and management are important pathways for enterprises to achieve their expected goals [22]. On the contrary, Song & Wu [23] and Wang et al. [24] contend that financialization involves being willing to assume more financial risks when facing operational and financial challenges. Over-financialization is deemed to occur when these risks surpass expectations, offering valuable insights for refining the theoretical boundaries of financialization and optimizing corporate governance practices.

Research on managerial myopia tends to focus on measurement methods, which can effectively reflect managers' behavioral tendencies. Shortsighted behavior exists in various fields, such as managerial shortsightedness, market shortsightedness, and investment shortsightedness [25–27]. Among them, the behavioral dimensions of managers have been extensively examined, with scholars contending that managers wield significant influence in strategy formulation and decision-making [28–30]. As the helmsman of the enterprise, managers play an important role in the strategic formulation and decision-making of the enterprise [29]. To

help the corporate shareholders and sectors of society recognize the shortsighted behavior of managers, some scholars have captured and managed shortsighted behavior by analyzing the word types and word frequencies used in the language of the experimental subjects [25, 31]. In one work, the shortsighted behavior is captured by analyzing the word types and word frequencies used in the language of the experimental subjects. The reason the text can be used to describe the characteristics of managers is as follows: First, text can effectively capture the characteristics of people. For example, the more emphasis there is on "past", "once", and similar words in a person's language, the more attention they pay to the past; the more emphasis there is on words such as "future", "possible", and "to go", the more they pay attention to the future [32]. Second, the characteristics of managers greatly affect the characteristics of corporate information disclosure [33]. Management discussion and analysis (MD&A), as a manager's review of the business situation during the reporting period, as well as an exposition of the opportunities, challenges, and risks faced by the next year's business plan and the future development of the company, can directly show the characteristics of managers. Li [34] reports that it is reliable to depict managers' traits through texts such as MD&A.

Regarding the impact of managerial myopia on corporate financialization, it can clearly be understood that managerial myopia is characterized by managers prioritizing short-term profit decisions at the potential cost of the company's long-term interests [8]. From the perspective of economic motivation, management, out of consideration for their own position, salary, and reputation, may use information asymmetry to choose some short-term investment schemes that can quickly generate returns, rather than making strategic decisions from the perspective of long-term best interests. For example, Gopalan et al. [35] reports that the shorter the average execution period of the management compensation contract, the easier it is for the management to make shortsighted behavior. Bolton [36] finds that shortsighted managers may sacrifice the long-term interests of enterprises to obtain excess compensation brought about by stock price fluctuations. Graham et al. [37] reveals that to obtain stable income and maintain their reputation, management may adopt some hidden profit manipulation methods and even make activities that sacrifice the long-term value of the enterprise. From the perspective of external pressure, the investment preference of short-term institutional investors [38], analyst tracking [25], and the number of financial report disclosures [39] may affect managerial shortsightedness. In this context, financialization emerges as a novel form of surplus management and a means of adjusting book profits, appealing to shortsighted managers and consequently intensifying the risk of over-financialization [40]. Although managers tend to prioritize growth and shareholders emphasize profits, the performance of financialization aligns with the shared preferences of both shortsighted managers and shareholders. This convergence of interests in the realm of financialization has led to interconnection of corporate financialization and managerial myopia [41].

Hence, a fundamental contradiction between micro-level enterprises and macroeconomic financialization is the widespread financialization of enterprises, in which the degree of financialization of enterprises does not match the level of their own resource management, resulting in the phenomenon of the crowding-out of enterprises' investment in fixed assets from "real to virtual". It must be admitted that financialization represents a kind of financial investment behavior driven by enterprise resource management, and enterprise financialization is also conducive to enterprise development to a certain extent. Therefore, compared with the financialization behavior caused by managers' short-sightedness, the study believes that it is more important to prevent the excessive financialization behavior caused by managers' short-sightedness. The main mechanism are shown in Fig 1, where corporate financialization decisions mainly influenced by the "reserve motives" and "profit motives". Among them, profit-motivated enterprises are subdivided into "investment substitution" and "investing surplus" types.

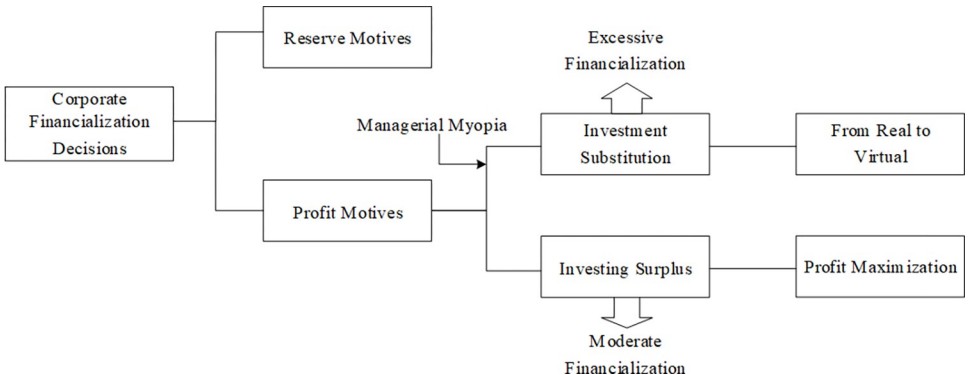

**Fig 1. Corporate financialization decision and excessive financialization.**

The excessive financialization measures taken by "investment substitution" enterprises formed under managerial myopia will cause "from real to virtual". While "investing surplus" will take moderate financialization measures to realize the profit maximization.

## Research methodology

### Operational definitions of research variables

**Over-financialization.** Certainly, clarifying the degree of corporate financialization and delineating the "moderation" boundary are foundational steps in the research process. This paper refers to the definition of Demir [42] to employ the ratio of financial assets to total assets as an index to measure financialization degree, where financial assets mainly include trading financial assets, sustainable sale financial assets, held-to-maturity investments, loans and advances, derivative financial instruments, long-term equity investments, and investment properties.

Consistent with Richardson's notion [43] of the non-efficiency investment index, we construct indicators for gauging the degree of over-financialization and the occurrence of financialization. The essence of financialization is the investment behavior decision-making of enterprises based on their available resources, which is mainly affected by the firm's size, age, cash flow status, financial leverage, growth potential, profitability, and historical financial asset allocation. In this way, this method differs from Wang et al.'s calculation of excessive financialization based on financial risks [24], as it is more conducive for enterprises to focus on long-term development prospects. Therefore, we construct Eq (1) to fit the optimal financialization level of enterprises:

$$Fin_{i,t} = \alpha_0 + \alpha_1 Fin_{i,t-1} + \alpha_2 Size_{i,t-1} + \alpha_3 Age_{i,t-1} + \alpha_4 Flow_{i,t-1} + \alpha_5 Lev_{i,t-1}$$
$$+ \alpha_6 Growth_{i,t-1} + \alpha_7 ROA_{i,t-1} + \vartheta_{firm} + \vartheta_{year} + \varepsilon_{i,t} \tag{1}$$

where $Fin_{i,t}$ is the degree of financialization of firm $i$ in period $t$; $Fin_{i,t-1}$ is the degree of financialization of firm $i$ in period $t$ (t-1); $Size_{i,t-1}$ is the size of the firm's capital in the previous period, measured using the logarithm of the firm's total; $Age_{i,t-1}$ is the logarization of the operating age of the enterprise for the previous period; $Flow_{i,t-1}$ is the cash flow position of the enterprise's operations in the previous period, which is derived from the total net cash flow as a percentage of total assets; $Lev_{i,t-1}$ is the previous financial leverage of the enterprise, measured by the asset-liability ratio $t$; x is the growth ability of the enterprise in the previous period, measured by the growth rate of operating income; $ROA_{i,t-1}$ is the prior period

profitability of the firm, where profitability is measured by prior period return on total assets. $\vartheta_{firm}$ and $\vartheta_{year}$ denote the firm fixed effect and year fixed effect, respectively. The paper uses the firm fixed effect instead of the industry fixed effect because whether a firm's financialization is excessively dependent on the state of its internal asset management, and fixing firms helps avoid the omitted variable problem.

Though Eq (1), the optimal financialization level of each entity enterprise can be fitted. $\varepsilon$, which is the degree of over-financialization *ExFin* in this research, is the distance between the actual financialization degree of the enterprise and the optimal. The positive residual indicates over-financialization of an enterprise, and the negative means that the firm is financialized within a moderate range. The larger residual the residual value is, the more likely the enterprise is over-financialized. In addition, we construct an indicator, *IExFin*, of whether there is over-financialization, based on the optimal financialization level. When *ExFin>0*, the enterprise has over-financialization, and *IExFin* is assigned to 1; when *IExFin* indicates that the enterprise finance is within a moderate range, *IExFin* is assigned to 0.

**Managerial myopia.** In the field of managerial myopia, the method of capturing managerial strategic information through word frequency text has become relatively mature. Brochet et al. [25] report on the word frequency ratio of "time-domain" vocabulary to effectively capture managerial myopia. Similarly, Hu et al. [31] build a comprehensive set of word frequency statistics tailored to the specific context of managerial myopia in Chinese enterprises owing to the nuances between Chinese and English dictionaries, and then they curate a selection of 43 words under the "shortsighted domain" category as managerial myopia indicators. This sophisticated dictionary-based method enables a nuanced understanding of managerial myopia within the Chinese business landscape.

The specific operation steps of measuring managerial myopia are as follows: (1) Convert to TXT (text) format: The managerial myopia index is converted into a TXT format utilizing Python based on Portable Document Format (PDF) annual reports of A-share listed companies in Shanghai and Shenzhen; (2) Extract the MD&A chapters, which often encapsulate critical insights into managerial strategies, in the annual financial reports; (3) Perform word frequency partitioning: The Jieba toolkit, a versatile Chinese language processing tool in Python, is employed to conduct word frequency partitioning on the annual report text of the listed companies. Then, the word frequency of "myopia field" is counted after filtering out irrelevant or deactivated words; (4) Calculate the total word frequency of MD&A chapters based on Python's Jieba toolkit; (5) Use the word frequency of "myopia field" as a percentage of the total frequency of MD&A and multiply it by 100 to obtain the indicator of managerial myopia index (*Myopia*). The larger the *Myopia* value, the more myopic the manager.

## Control variables

Financialization is the investment decision of enterprises based on resource endowment, which means that the current level of financialization reflects the ongoing enterprises' adjustments in resource allocation. In addition to investigating managerial myopia, we incorporate several control variables to perform a comprehensive analysis. These variables include 1) firm size: represented as the natural logarithm of total assets at the year-end; 2) firm age: expressed as the natural logarithm of the firm's operating years; 3) firm growth opportunity: measured by the growth rate of the year-end operating income; 4) firm debt ratio: defined as the ratio of total liabilities to total assets at the year-end; 5) firm liquidity: calculated as the ratio of the company's monetary assets to total assets at the year-end; 6) firm net interest rate: captured as the natural logarithm of the firm's net profit at the year-end. These variables collectively shed light on various aspects of the enterprise's operations, contributing to a more nuanced

understanding of how managerial decision-making and financialization interplay within the broader business context.

Taking into account the multifaceted nature of capturing managerial myopia within corporate annual reports and the collaborative nature of determining firms' financialization levels involving managers and shareholders, we incorporate a range of variables to ensure a comprehensive analysis. These variables, which serve as control measures, account for various dimensions of decision-making and governance within the enterprise: 1) equity market value: represented by the natural logarithm of the total market value of the enterprise's stock market, capturing the market perception and valuation of the company's worth; 2) Director-Cum-CEO: a binary indicator variable, taking the value of 1 when the CEO concurrently holds the position of the chairman of the board of directors and 0 otherwise. This variable accounts for the potential concentration of decision-making power; 3) proportion of independent directors: calculated as the ratio of independent directors to the total number of directors, reflecting the extent to which external perspectives influence governance; 4) ownership concentration: represented as the proportion of the top 10 shareholders' collective ownership in the company's total shares. Moreover, firm-specific fixed effects and time-specific fixed effects are introduced to mitigate the impact of unobserved or omitted variables that can potentially confound the results.

## Regression model of research

We employ an econometric model to test the causal relationship between managerial myopia and enterprise over-financialization. The research delves into two components of managerial myopia: its impact on the degree of over-financialization and its potential role in causing over-financialization. Importantly, the study distinguishes between the continuous variable representing the degree of over-financialization transformation and the binary variable indicating the presence or absence of over-financialization. As a result, different model settings are applied for these two components to effectively capture the nuances of the relationship.

First, the fixed-effect mixed OLS model (2) is set to test the impact of managerial myopia on the degree of over-financialization, as follows:

$$ExFin_{i,t} = \alpha_0 + \alpha_1 Myopia_{i,t} + \alpha_2 X_{i,t} + \tau + \varepsilon_{i,t} \tag{2}$$

where $ExFin$ is the dependent variable of Eq (2), indicating the degree of excessive financialization of company $i$ in time period $t$. $Myopia$ is the core independent variable, indicating the degree of managerial myopia in enterprise $i$ in time period $t$. $X$ is a set of control variables, including the enterprise's size, age, growth opportunities, debt ratio, liquidity, net interest rate, equity market value, directors and CEOs, proportion of independent directors, equity concentration, and other variables. $\tau$ is the fixed effect, including firm fixed effects and year fixed effects. $\varepsilon$ is the residual value.

Second, a panel binary choice of logistic Eq (3) is set to test the effect of managerial myopia on whether firms are engaging in over-financialization, as follows:

$$IExFin_{i,t} = P_{i,j,t} = \beta_0 + \beta_1 Myopia_{i,t} + \beta_2 X_{i,t} + \tau + \varepsilon_{i,t} \tag{3}$$

where $IExFin$ is the latent variable of Eq (3), denoting the probability $P_{i,j,t}$ of whether firm $i$ is in over-financialization at time period $t$; the remaining variables are consistent with the above. Then, Eq (3) can be transformed as

$$P(IExFin_{i,t} = 1 | x_{i,t}, \beta, \tau) = P(\varepsilon_{i,t} < \tau + x'_{i,t}\beta | x_{i,t}, \beta, \tau)$$
$$= F(\tau + x'_{i,t}\beta) \tag{4}$$

where $F(\cdot)$ is the cumulative distribution function of $\varepsilon_{i,t}$, and assuming that $\varepsilon_{i,t}$ obeys the density function of the logistic distribution, Eq (4) can be varied as

$$P(IExFin_{i,t} = 1|x_{i,t}, \beta, \tau) = \Lambda(\tau + x'_{i,t}\beta) = \frac{e^{\tau + x'_{i,t}\beta}}{1 + e^{\tau + x'_{i,t}\beta}} \tag{5}$$

## Sample selection

We focus on a comprehensive sample of listed companies in China from 2005 to 2022. To ensure the robustness and accuracy of the empirical analysis, we treat the sample as follows: (1) textual analysis inclusion: The research primarily employs textual analysis to capture managerial myopia characteristics, eliminating unpublished or discontinuous annual reports to ensure the completeness of the explanatory variable data; (2) financialization focus: The research specifically investigates the financialization behavior of real enterprises, excluding the samples of financial firms such as banks, securities, insurance and trusts; (3) exclusion of specific types: Companies categorized as *ST, ST and PT types are excluded from the sample to avoid outliers in the sample; (4) Missing variable removal: Samples with missing relevant variables in the relevant financial statements are removed from the analysis; (5) asset loading ratio threshold: Samples with asset loading ratios exceeding 100% and exclude financially abnormal samples are eliminated from the sample; (6) winsorization: To mitigate the impact of extreme values, we winsorize all continuous variables at the 1% and 99% levels. Following these rigorous criteria, the final total sample comprises 14,870 observations for unbalanced panel data. Table 1 reports the final sample distribution by industry, where Code J is the missing identifier for the financial industry in the Guidelines for Industry Classification of Listed Companies. Sample firms are mainly in the manufacturing sector, consistent with the industry distribution of listed companies in China, and the total number of Code C samples is 10,584 observations, accounting for 71.18%. The data regarding managerial myopia are extracted from the annual reports of A-share listed companies in the Shanghai and Shenzhen stock markets, and the

**Table 1. Sample distribution.**

| Code | Industry | Number | Percentage |
|---|---|---|---|
| A | Agriculture, forestry, livestock farming, fishery | 207 | 1.39% |
| B | Mining | 192 | 1.29% |
| C | Manufacturing | 10,584 | 71.18% |
| D | Utilities | 179 | 1.2% |
| E | Construction | 409 | 2.75% |
| F | Wholesale and retail | 752 | 5.06% |
| G | Transportation | 298 | 2% |
| H | Hotel and catering industry | 37 | 0.25% |
| I | Information transmission, software, and IT service | 954 | 6.42% |
| K | Real estate | 526 | 3.54% |
| L | Leasing and commerce service | 107 | 0.72% |
| M | Scientific research and technology service | 125 | 0.84% |
| N | Water conservancy, environment, and public facilities | 146 | 0.98% |
| O | Resident services, repairs, and other services | 8 | 0.05% |
| P | Education | 2 | 0.01% |
| Q | Public health and social work | 29 | 0.2% |
| R | Culture, sports, and entertainment | 121 | 0.81% |
| S | Comprehensive | 194 | 1.3% |
| | Total | 14870 | 100% |

Python program is employed to perform web crawling and compile relevant word frequency statistics. Additional variable data are derived from the CSMAR database, ensuring a comprehensive, robust dataset for the empirical analysis.

## Empirical analysis

**Descriptive statistics.** Descriptive statistics of research variables are shown in Table 2, focusing on the selected listed entity firms in China from 2005 to 2022, yielding a total of 14,870 observations after processing. Table 1 reports the descriptive statistics of all variables. It can be found that the degree of Chinese enterprises' over-financialization has a normal distribution, with the mean value around 0. The range of values spans from a minimum of −31.01 to a maximum of 55.64. At the level of whether or not over-financialization exists, a considerable portion of enterprise samples have over-financialization in the sample as a whole, with the mean value being 0.34, and more than 60% of enterprises being in a rational financialization state. In the descriptive statistics of the core independent variables, Chinese enterprises have different degrees of managerial myopia. The proportion of "managerial myopia" within the total word frequency of MD&A in the annual reports is within in the range of [0.01%, 1.96%], and the average value of manager myopia is observed at approximately 0.23% across the entire sample.

At the level of enterprise resource control, the asset sizes of Chinese listed entity enterprises do not differ much after eliminating the enterprises with abnormal debt ratio. Most enterprises maintain a high level of business growth, with corporate liquidity and net profit showing a normal distribution. At the management control level, the average value of director-cum-CEO in Chinese enterprises is 0.26, indicating that a quarter of the board of directors in the sample directly manages the company. The mean proportion of independent directors in the board of directors is 37.10%, and the vast majority of enterprises comply with at least one-third of the provisions of the independent director system for independent directors. The average concentration of the top 10 shareholders of each sample is 10%, but substantial variation exists among enterprises, with the largest company demonstrating an equity concentration of as high as 64.29%.

## Tests related to the classical hypothesis of regression

Table 3 presents the outcomes of the model tests. In this section, the variance inflation factor (VIF) test, F-Limer test, and Hausman test are performed on the fixed-effects model and the

**Table 2. Descriptive statistics.**

| Variable | Symbols | Obs | Mean | Std.Dev | Min | Max |
|---|---|---|---|---|---|---|
| Excessive Financialization | *ExFin* | 14870 | -0.0292 | 4.8981 | -31.0092 | 55.6388 |
| Whether Excessive Financialization | *IExFin* | 14870 | 0.3389 | 0.4734 | 0.0000 | 1.0000 |
| Managerial Myopia | *Myopia* | 14870 | 0.2338 | 0.1514 | 0.0108 | 1.9608 |
| Firm's Age | *Age* | 14870 | 2.8731 | 0.3392 | 1.3863 | 4.1744 |
| Firm's Growth | *Growth* | 14870 | 0.3340 | 0.7740 | -0.7027 | 8.7011 |
| Firm Debt Ratio | *Lev* | 14870 | 0.4201 | 0.1654 | 0.0701 | 0.8444 |
| Firm's Flunency | *Flow* | 14870 | 0.0098 | 0.0147 | -0.0906 | 0.1047 |
| Firm's Size | *Size* | 14870 | 22.2678 | 1.3314 | 19.6158 | 29.3458 |
| Return on Assets | *ROA* | 14870 | 0.0388 | 0.0478 | -0.2070 | 0.1954 |
| Equity Market Value | *Tobin* | 14870 | 22.2724 | 0.9376 | 19.8202 | 25.4168 |
| Director-cum-CEO | *Twoduty* | 14870 | 0.2564 | 0.4367 | 0.0000 | 1.0000 |
| Independent Directors Proportion | *Manage* | 14870 | 37.0970 | 5.0030 | 22.2200 | 57.1400 |
| Shareholding Concentration | *Share* | 14870 | 10.3940 | 16.2097 | 0.0000 | 64.2878 |

**Table 3. The results of the regression hypothesis test.**

| Test | Test Statistic | Significance Level | Test Result |
|---|---|---|---|
| VIF of regression | 1.30 | - | No multicollinearity |
| VIF of Logit | 1.30 | - | No multicollinearity |
| F Limer of regression | 3.49 | 0.0000 | Admitting panel data pattern |
| F Limer of logistics | 54.71 | 0.0000 | Admitting panel data pattern |
| Hausman of regression | 161.96 | 0.0000 | Fixed effects of accept |
| Hausman of logistics | 154.97 | 0.0000 | Fixed effects of accept |

logistics model. The results show no multicollinearity exists between the variables, as evidenced by a VIF value of 1.3. This implies that the examined variables are not highly correlated, thereby supporting the reliability of the model. The significance levels of the F-Limer test for the two models are both less than 5%, which indicates that the panel data model is valid and accepted. The Hausman test also yields significance levels below 5%, and the fixed-effects model setting is deemed suitable and accepted.

## Result of the research hypothesis test

The regression results of the effect of managerial myopia on corporate over-financialization are shown in Table 4. Column 1 shows the linear regression results of managerial myopia on the degree of corporate over-financialization, and Column (2) presents whether managerial myopia triggers corporate over-financialization. Column (1) shows that the coefficient of *Myopia* to *ExFin* is positive and significant ($p < 0.05$). This means that the degree of corporate over-financialization increases by 11.24% (0.7424 x 0.1514) for every unit of standard deviation increase in managerial myopia. Moreover, the column (2) results do not show the constant term with R-squared owing to the fixed panel logistic model used in the test. After 1,442 groups (5,750 obs) of firm samples are eliminated in the panel logistic regression because these groups have either positive or negative outcomes, the coefficient of *Myopia* to *ExFin* remains positive and significant ($p < 0.05$), and the probability of over-financialization increases by 7.01% (0.4633 x 0.1514) when managerial myopia increases by one standard deviation. The results indicate that managerial myopia manifests in short-termism, leading to increased financial asset allocation beyond the reasonable level of resource allocation for enterprises, which is detrimental to long-term development.

## Robustness check

To enhance the robustness of the empirical results, we conduct various sensitivity analyses involving adjustments to the sample interval, the replacement of variables, and endogeneity testing. The outcomes of these analyses are presented in Table 5 (adjusting sample interval and replacement variables) and Table 6 (endogeneity test results).

## Eliminating the impact of the financial crisis

The root causes of the 2008 global financial crisis stemmed from improper real estate financial policies and the misuse of financial derivatives, which clearly fueled excessive financialization behavior in enterprises. After the financial crisis of 2008, some enterprises learned from experience, leading to a moderation in the situation of excessive financialization. Hence, this part excludes the impact of the 2008 global financial crisis and adjusts the study sample to the period from 2009 to 2022. Columns (1) and (2) of Table 5 are the effect managerial myopia on the degree of over-financialization and whether over-financialization occurs. The results show

**Table 4. Estimation result: Effects of managerial myopia on excessive financialization.**

|  | (1) | (2) |
|---|---|---|
|  | *ExFin* | *IExFin* |
| *Myopia* | 0.7424** | 0.4633** |
|  | (0.2900) | (0.2174) |
| *Age* | 2.1511*** | 1.8564*** |
|  | (0.6715) | (0.4762) |
| *Growth* | 0.0373 | 0.0670* |
|  | (0.0592) | (0.0394) |
| *Lev* | -1.7442*** | 0.2841 |
|  | (0.5278) | (0.2994) |
| *Flow* | 2.2357 | 2.7582 |
|  | (2.9354) | (2.0949) |
| *Size* | -0.2151*** | -0.1375*** |
|  | (0.0515) | (0.0385) |
| *ROA* | -2.4580** | -1.1636 |
|  | (1.0853) | (0.7539) |
| *Tobin* | 0.0527 | -0.0644 |
|  | (0.1024) | (0.0680) |
| *Twoduty* | -0.0588 | 0.0418 |
|  | (0.1235) | (0.0847) |
| *Manage* | 0.0090 | -0.0062 |
|  | (0.0123) | (0.0089) |
| *Share* | -0.0080 | -0.0094** |
|  | (0.0060) | (0.0048) |
| *_Cons* | -1.3236 | - |
|  | (2.6335) |  |
| *Company* | YES | YES |
| *year* | YES | YES |
| N | 14,870 | 9,120 |
| Adj R$^2$ | 0.5675 | - |

Note: Standard errors are in brackets

***,**,* indicate significant at 1%, 5%, and 10% level of significance, respectively.

that the coefficient of *Myopia* is positively significant (p<0.10). The results indicate that even after excluding the influence of financial crises, managerial myopia still promotes excessive financialization in enterprises. The conclusion is consistent with the previous section, meaning that the conclusions are robust.

## Replacement of variable measure

The substitution of explanatory variables and dependent variables is a commonly used method in robustness testing, aiming to examine whether the causal relationship between variables still holds. The degree of financialization, *Fin*, is used to replace the over-financialization index in this part, and the results are shown in column of Table 5. The coefficient of *Myopia* remains positive and significant (p<0.10). The conclusion that managerial myopia increases financial asset allocation is consistent with the findings of [44]. Differences exist in the explanatory significance between the two. Managerial myopia increases the degree of enterprises' financialization, while the increase in the degree of financialization may be conducive to the optimal

**Table 5. Robustness tests.**

| | Eliminating the Impact of the Financial Crisis | | Replacement of variable measure | | |
|---|---|---|---|---|---|
| | (1) | (2) | (3) | (4) | (5) |
| | *ExFin* | *IExFin* | *Fin* | *ExFin* | *IExFin* |
| *Myopia* | 0.5608* | 0.4239* | 0.6801* | | |
| | (0.2968) | (0.2567) | (0.3954) | | |
| *ART* | | | | 0.0007*** | 0.0003*** |
| | | | | (0.0002) | (0.0001) |
| *Age* | 3.7258*** | 3.6243*** | 5.5976*** | 2.0787*** | 1.8156*** |
| | (0.7055) | (0.6181) | (0.9627) | (0.6691) | (0.4762) |
| *Growth* | 0.0265 | 0.0523 | 0.1080 | 0.0375 | 0.0677* |
| | (0.0585) | (0.0445) | (0.0739) | (0.0592) | (0.0394) |
| *Lev* | -1.2123** | 0.2256 | -7.1217*** | -1.8373*** | 0.2420 |
| | (0.5294) | (0.3443) | (0.6904) | (0.5285) | (0.2998) |
| *Flow* | 3.3556 | 3.7221 | -14.6511*** | 2.1213 | 2.6987 |
| | (2.9989) | (2.3534) | (3.8230) | (2.9307) | (2.0947) |
| *Size* | -0.2304*** | -0.1516*** | -0.4126*** | -0.1893*** | -0.1216*** |
| | (0.0507) | (0.0401) | (0.0703) | (0.0516) | (0.0387) |
| *ROA* | -2.1894** | -1.3006 | -5.1299*** | -2.4015** | -1.1286 |
| | (1.0872) | (0.8214) | (1.4659) | (1.0814) | (0.7551) |
| *Tobin* | 0.0181 | -0.0440 | -0.1060 | -0.0173 | -0.1039 |
| | (0.0984) | (0.0773) | (0.1373) | (0.1027) | (0.0693) |
| *Twoduty* | -0.0188 | 0.1259 | 0.0177 | -0.0141 | 0.0733 |
| | (0.1299) | (0.0921) | (0.1660) | (0.1219) | (0.0839) |
| *Manage* | 0.0244* | 0.0044 | 0.0076 | 0.0081 | -0.0063 |
| | (0.0134) | (0.0106) | (0.0164) | (0.0125) | (0.0090) |
| *Share* | -0.0054 | -0.0057 | -0.0373*** | -0.0073 | -0.0088* |
| | (0.0061) | (0.0050) | (0.0079) | (0.0060) | (0.0048) |
| *_Cons* | -7.1693** | - | 6.3070* | -0.4815 | - |
| | (2.9559) | | (3.7153) | (2.7804) | |
| *Company* | YES | YES | YES | YES | YES |
| *year* | YES | YES | YES | YES | YES |
| Obs. | 13,445 | 7,460 | 14,870 | 14,870 | 9,120 |
| R-squared | 0.6046 | - | 0.7432 | 0.5679 | - |

Note: Standard errors are in brackets

***,**,* indicate significant at 1%, 5%, and 10% level of significance, respectively.

financialization of firms. Subsequently, this part uses the stock turnover rate to replace the indicator of managerial myopia. This is because managers largely adopt short-term behaviors to improve market valuation to cater to investors, and these short-term behaviors increase the stock turnover rate of enterprises. The result that the coefficient of *ART* is positively significant (p<0.05) further confirms this idea.

## Endogeneity test

Causal inference should exclude bidirectional causality between independent and dependent variables, meaning that short-term gains from corporate financialization may lead to managerial overconfidence, making them more myopic. Therefore, the paper chose the instrumental variable approach to deal with endogeneity, and different instrumental variable (IV) analysis

models are selected based on distinction between the over-financialization degree and the indicators of whether over-financialization is present. The traditional two-stage least squares (2SLS) test is employed for the endogenous test of the former over-financialization degree. Since no instrumental variable method model exists for fixed Logit, the alternative model test (IV-Probit model test) is used for the latter index.

For the selection of instrumental variables, we choose whether the manager has experienced famine as an instrumental variable. China refers to 1959–1961 as the "Three-year Difficult Period" or "Three-year Natural Disasters." During this period, China's farmlands suffered from large-scale natural disasters for several years in a row, facing a nationwide food shortage crisis with about 2.5 million deaths due to starvation. Managers' early life experiences tend to influence their corporate decision-making. When managers have experienced famine in their early years, they tend to be conservative in decision-making on whether to over-financialize businesses, and they set aside part of the capital to cope with the "famine." Therefore, the managerial age of entrepreneurs born before 1959 is set to 1 in this research and set to 0 for those born after 1959.

In summary, the results of the endogeneity tests are shown comprehensively in Table 6. Columns (1) and (3) are the first-stage results of the two instrumental variables methods, while Columns (2) and (4) are the second-stage results. The findings indicate that the instrumental variables used in the analysis exhibit positive and statistically significant (p<0.01). Moreover, the coefficient of managerial myopia remains positively significant in the second stage, reinforcing the robustness of the observed relationship. The Anderson canon. corr. Lagrange Multiplicator (LM) statistic and Cragg–Donald Wald F (joint hypotheses) statistic also pass the test in the research, suggesting that the instrumental variables do not face issues of overidentification or weak instrumental variables.

## Moderating role of financial constraints and financial distress

The subsequent subsection delves into the potential "reservoir effect" of corporate financialization in the context of mitigating financial risks. Specifically, this subsection mainly examines whether the presence of financial risks amplifies the likelihood of over-financialization. The research mainly classifies financial risks into two categories: financing constraints and financial distress. Financing constraints entail challenges faced by enterprises in raising external funds, while financial distress denotes a financial crisis that can disrupt the capital turnover process. This paper employs the size and age index (SA index) and zeta score (Z-Score) to quantify these two risks. The results are shown in Table 7.

Columns (1) and (3) present the outcomes of assessing the moderating influence of the two risk variables on the relationship between managerial myopia and the degree of over-financialization. Columns (2) and (4) are the moderating role of the two variables in the influence of managerial myopia on the degree of whether over-financialization. The results reveal that the coefficient *SA×Myopia* is positive in both Columns (1) and (2), but that it is statistically significant only in Columns (1) (p<0.05). Similarly, the coefficient *Zfin×Myopia* is positive in both Columns (3) and (4), with statistical significance observed only in Columns (3) (p<0.10). These findings suggest that an increase in financial risk, as indicated by the SA index and Z-Score, intensifies the degree of financialization driven by managerial myopia. However, this heightened financial risk does not necessarily trigger over-financialization among enterprises.

## Conclusion

This paper has constructed the optimal financialization level index of enterprises based on the sample of nonfinancial listed companies from 2005 to 2022 in China, and has empirically

**Table 6. Robustness tests.**

| | (1) | (2) | (3) | (4) |
|---|---|---|---|---|
| | *Myopia* | *ExFin* | *Myopia* | *IExFin* |
| Famine | 0.0372*** | | 0.0197*** | |
| | (0.0055) | | (0.0038) | |
| Myopia | | 9.1071* | | 5.8601*** |
| | | (5.0112) | | (0.1045) |
| Age | 0.0421*** | 0.7289*** | 0.0286*** | 0.1160*** |
| | (0.0045) | (0.2625) | (0.0043) | (0.0285) |
| Growth | -0.0043*** | 0.0124 | -0.0090*** | 0.0852*** |
| | (0.0016) | (0.0597) | (0.0016) | (0.0121) |
| Lev | -0.0086 | -2.0051*** | -0.0122 | -0.0443 |
| | (0.0088) | (0.3010) | (0.0084) | (0.0629) |
| Flow | 0.2523*** | -12.3461*** | 0.3194*** | -3.2404*** |
| | (0.0887) | (3.2744) | (0.0882) | (0.6832) |
| Size | -0.0005 | -0.1070** | -0.0006 | -0.0072 |
| | (0.0015) | (0.0497) | (0.0013) | (0.0091) |
| ROA | -0.1281*** | -0.5411 | -0.1002*** | 0.2986 |
| | (0.0296) | (1.2023) | (0.0294) | (0.2239) |
| Tobin | 0.0007 | 0.3150*** | -0.0016 | 0.0615*** |
| | (0.0020) | (0.0691) | (0.0019) | (0.0129) |
| Twoduty | -0.0173*** | -0.0187 | -0.0197*** | 0.1361*** |
| | (0.0027) | (0.1271) | (0.0028) | (0.0217) |
| Manage | -0.0005** | 0.0316*** | -0.0007*** | 0.0082*** |
| | (0.0002) | (0.0083) | (0.0002) | (0.0018) |
| Share | -0.0011*** | -0.0024 | -0.0013*** | 0.0070*** |
| | (0.0001) | (0.0061) | (0.0001) | (0.0007) |
| _Cons | 0.3878*** | -8.9623*** | 0.3672*** | -3.5539*** |
| | (0.0401) | (2.0352) | (0.0367) | (0.2520) |
| Company | YES | YES | YES | YES |
| year | YES | YES | YES | YES |
| Anderson canon. corr. LM statistic | 48.41 | | - | |
| | (0.00) | | | |
| Cragg-Donald Wald F statistic | 48.22 | | 601.64 | |
| | [0.000] | | [0.000] | |
| Obs. | 14870 | 14870 | 14870 | 14870 |
| R-squared | 0.1543 | 0.0738 | 0.1543 | - |

Note: Standard errors are in brackets

***,**,* indicate significant at 1%, 5%, and 10% level of significance, respectively.

analyzed the impact of managerial myopia on over-financialization of firms on this index basis. The findings reveal a prevalent trend of financialization in China's real enterprises. While a significant number of firms demonstrate behaviors of over-financialization, the majority of enterprises fall within a moderate range of financialization practices. From the perspective of managers in corporate governance, managerial myopia favors financialization to obtain short-term benefits and triggers over-financialization behaviors, which are detrimental to firms' long-term interests. Under financial distress and financing constraints, such behavior exacerbates the shortsightedness of managers in increasing their holdings of financial assets to make quick short-term gains to tide over difficulties. However, this behavior does not

**Table 7. Estimation result: Moderating effects of financial risk.**

| | (1) | (2) | (3) | (4) |
|---|---|---|---|---|
| | *ExFin* | *IExFin* | *ExFin* | *IExFin* |
| Myopia | 12.1332** | 1.5046 | 0.2329 | 0.3888 |
| | (4.7101) | (3.1924) | (0.4102) | (0.2993) |
| SA | -1.8289* | -1.6936** | | |
| | (0.9767) | (0.7026) | | |
| SA×Myopia | 3.0014** | 0.2692 | | |
| | (1.2370) | (0.8399) | | |
| Zfin | | | -0.0744*** | -0.0252* |
| | | | (0.0215) | (0.0143) |
| Zfin×Myopia | | | 0.1218* | 0.0205 |
| | | | (0.0740) | (0.0490) |
| Age | 2.1254*** | 1.7924*** | 2.0203*** | 1.7959*** |
| | (0.6751) | (0.4784) | (0.6698) | (0.4771) |
| Growth | 0.0390 | 0.0683* | 0.0384 | 0.0671* |
| | (0.0593) | (0.0395) | (0.0589) | (0.0394) |
| Lev | -1.8390*** | 0.2317 | -2.2850*** | 0.0382 |
| | (0.5336) | (0.3009) | (0.5602) | (0.3198) |
| Flow | 2.3136 | 2.8526 | 2.4789 | 2.8758 |
| | (2.9320) | (2.0961) | (2.9358) | (2.0977) |
| Size | -0.2200*** | -0.1363*** | -0.2456*** | -0.1528*** |
| | (0.0515) | (0.0384) | (0.0517) | (0.0392) |
| ROA | -2.5219** | -1.2858* | -2.0858* | -1.0165 |
| | (1.0852) | (0.7563) | (1.0825) | (0.7572) |
| Tobin | 0.0396 | -0.0695 | 0.1659 | -0.0145 |
| | (0.1015) | (0.0682) | (0.1080) | (0.0716) |
| Twoduty | -0.0114 | 0.0809 | -0.0139 | 0.0788 |
| | (0.1222) | (0.0839) | (0.1220) | (0.0839) |
| Manage | 0.0095 | -0.0059 | 0.0093 | -0.0058 |
| | (0.0125) | (0.0090) | (0.0125) | (0.0090) |
| Share | -0.0064 | -0.0084* | -0.0083 | -0.0097** |
| | (0.0060) | (0.0048) | (0.0060) | (0.0048) |
| _Cons | -7.8775* | - | -2.5366 | - |
| | (4.4167) | | (2.8342) | |
| Compay | YES | YES | YES | YES |
| year | YES | YES | YES | YES |
| Obs. | 14,870 | 9,120 | 14,870 | 9,120 |
| R-squared | 0.5678 | - | 0.5681 | - |

Note: Standard errors are in brackets

***,**,* indicate significant at 1%, 5%, and 10% level of significance, respectively.

necessarily result in over-financialization, suggesting that financialization under such circumstances may be a strategic response, rather than a cause of over-financialization.

## Discussion and suggestions

This paper has introduced a valuable distinction between over-financialization and corporate financialization, which contributes to a more rational understanding and analysis of real firms'

financialization behaviors, and it has highlighted the difference between managerial myopia and financialization. The findings suggest that managerial myopia leads to corporate over-financialization, providing a new explanation for the intrinsic motivation. Furthermore, the findings highlight a potential avenue for corporate governance strategies to address and mitigate the influence of managerial myopia, thereby curbing the occurrence of over-financialization. Finally, the findings suggest that researchers can explore the broader economic ramifications of excessive finance from the over-financialization perspective in the future, rather than simply viewing financialization as a homogenous behavior.

## Research limitations

While this study contributes significantly to the understanding of corporate financialization and proposes a more balanced governance approach for firms by considering both short-term and long-term shareholder interests, it does have certain limitations. From the long-term governance perspective, companies that spend more earnings on innovation and infrastructure may be more likely to achieve higher levels of succuss in the future. In contrast, the prevalence of financialization behaviors in the sample of real firms in China can hinder direct comparisons between the two various approaches, which is an inherent constraint of this research. In light of these limitations, future research can conduct delve deeper into the examination of the long-term earnings and performance outcomes of financialized firms versus not financialized firms.

## Acknowledgments

We would like to thank all members of the Doctoral Program in Collaborative Innovation Center of Modern Grain Circulation and Safety, and all support from the Nanjing University of Finance and Economics for making it possible to carry out this work.

## Author Contributions

**Conceptualization:** Ju Ye.

**Data curation:** Yanchi Chen, Ju Ye, Qi Shi.

**Formal analysis:** Yanchi Chen, Ju Ye.

**Funding acquisition:** Yanchi Chen, Ju Ye, Qi Shi.

**Methodology:** Ju Ye.

**Project administration:** Qi Shi.

**Supervision:** Yanchi Chen, Ju Ye.

**Validation:** Yanchi Chen, Ju Ye.

**Visualization:** Yanchi Chen, Ju Ye, Qi Shi.

**Writing – original draft:** Yanchi Chen, Ju Ye, Qi Shi.

**Writing – review & editing:** Yanchi Chen, Ju Ye, Qi Shi.

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
