## [Decision Letter · Decision Letter 0]

9 Apr 2024

PONE-D-24-06837Does managerial myopia promote enterprises over-financialization? Evidence from listed firms in ChinaPLOS ONE

Dear Dr. Ye,

Thank you for submitting your manuscript to PLOS ONE. After careful consideration, we feel that it has merit but does not fully meet PLOS ONE’s publication criteria as it currently stands. Therefore, we invite you to submit a revised version of the manuscript that addresses the points raised during the review process.

Please submit your revised manuscript by May 24 2024 11:59PM. If you will need more time than this to complete your revisions, please reply to this message or contact the journal office at plosone@plos.org. Please include the following items when submitting your revised manuscript:A rebuttal letter that responds to each point raised by the academic editor and reviewer(s). You should upload this letter as a separate file labeled 'Response to Reviewers'.A marked-up copy of your manuscript that highlights changes made to the original version. You should upload this as a separate file labeled 'Revised Manuscript with Track Changes'.An unmarked version of your revised paper without tracked changes. You should upload this as a separate file labeled 'Manuscript'.

We look forward to receiving your revised manuscript.

Kind regards,

Wajid Khan

Academic Editor

PLOS ONE

Journal Requirements:

   "This work was supported by the [National Social Science Fund of China] under Grant [22VRC007]; [Institute of Food and Strategic Reserves, Nanjing University of Finance and Economics] under Grant [ BSZX2023-07]."

Reviewers' comments:

Reviewer's Responses to Questions

**Comments to the Author**

1. Is the manuscript technically sound, and do the data support the conclusions?

Reviewer #1: Yes

Reviewer #2: Yes

2. Has the statistical analysis been performed appropriately and rigorously? 

Reviewer #1: Yes

Reviewer #2: Yes

3. Have the authors made all data underlying the findings in their manuscript fully available?

Reviewer #1: Yes

Reviewer #2: Yes

4. Is the manuscript presented in an intelligible fashion and written in standard English?

Reviewer #1: Yes

Reviewer #2: Yes

5. Review Comments to the Author

Reviewer #1: This paper investigates whether managerial myopia promotes over-financialization in enterprises. The theme of this paper is interesting. However, it needs to be amended from the following points to meet the requirements of publication.

1. Literature review section needs optimization, lacks research on the impact of managerial myopia on over-financialization in enterprises.

2.It is suggested to summarize the contribution of the article and placed it in the first part.

3. Please read the entire article carefully and correct some details. For example, the fonts in Table 4 are not consistent.

4. The logical hierarchy of the article's structure is somewhat redundant and needs adjustment. Both the literature review in the second section and the research methodology in the third section provide definitions of "managerial myopia" and "over-financialization in enterprises." It is suggested to consolidate and optimize these definitions.

5. The robustness check section lacks textual description, failing to elucidate how the results of the robustness check specifically contribute.

6. Other papers related to green low-carbon endogenous economic growth can refer to:

Yueh-Hsia Huang, Lan Sun, Tyng-Bin Ger. An analysis of enterprise resource planning systems and key determinants using the Delphi method and an analytic hierarchy process[J]. Data Science in Finance and Economics, 2023, 3(2): 166-183. doi: 10.3934/DSFE.2023010

Li, Z., Chen, H., & Mo, B. (2023). Can digital finance promote urban innovation? Evidence from China. Borsa Istanbul Review, 23(2), 285-296. doi:10.1016/j.bir.2022.10.006

Yonghong Zhong, Junhao Zhong. The spread of debt risk from real estate companies to banks: Evidence from China[J]. Quantitative Finance and Economics, 2023, 7(3): 371-390. doi: 10.3934/QFE.2023018

Reviewer #2: This topic deals with an interesting issue related to the assessment of the Research on the Does managerial myopia promote enterprises over-financialization? Evidence from listed firms in China: the comments are as follows

1.The authors should provide a clear explanation of the study's scope and objectives, substantiated by relevant studies. Additionally, the literature review appears weak and lacks cohesion.

2.The measurement methods should be justified and compared to other methods to explain why this specific method was chosen.

3.The author should also justify the choice of statistical methods, sample size, and the application of the chosen tests to ensure their validity and correctness.

4.The author(s) should explain whether the study is in line with prior studies or not, also whether it supports any theories.

6. PLOS authors have the option to publish the peer review history of their article (what does this mean?). If published, this will include your full peer review and any attached files.

Reviewer #1: No

Reviewer #2: No

---

## [Author Response · Author response to Decision Letter 0]

15 May 2024

Dear specific reviewer and editor:

On behalf of all co-authors, I would like to thank you and the reviewers for your serious and thorough work in refereeing this paper. Thank you for giving us a chance to revise and improve the quality of our article.

We have read the reviewers’ and your comments carefully and have made revision which marked in red in the paper. We have tried our best to revise our manuscript according to all suggestions.

Attached please find the revised version, which we hope that you will find this updated manuscript to your satisfaction and consider it for publication as an article in PLOS ONE. A point-by-point response to the reviewers’ comments and concerns is given below.

Thank you for taking the time to consider our research and we look forward to hearing from you at your earliest convenience

Yours sincerely,

Corresponding author

Ju YE

---

## [Decision Letter · Decision Letter 1]

7 Aug 2024

Does managerial myopia promote enterprises over-financialization? Evidence from listed firms in China

PONE-D-24-06837R1

Dear Dr. Ye,

We’re pleased to inform you that your manuscript has been judged scientifically suitable for publication and will be formally accepted for publication once it meets all outstanding technical requirements.

Kind regards,

Wajid Khan

Academic Editor

PLOS ONE

Additional Editor Comments (optional):

Reviewers' comments:

Reviewer's Responses to Questions

**Comments to the Author**

1. If the authors have adequately addressed your comments raised in a previous round of review and you feel that this manuscript is now acceptable for publication, you may indicate that here to bypass the “Comments to the Author” section, enter your conflict of interest statement in the “Confidential to Editor” section, and submit your "Accept" recommendation.

Reviewer #2: All comments have been addressed

2. Is the manuscript technically sound, and do the data support the conclusions?

Reviewer #2: Yes

3. Has the statistical analysis been performed appropriately and rigorously? 

Reviewer #2: Yes

4. Have the authors made all data underlying the findings in their manuscript fully available?

Reviewer #2: Yes

5. Is the manuscript presented in an intelligible fashion and written in standard English?

Reviewer #2: Yes

6. Review Comments to the Author

Reviewer #2: (No Response)

7. PLOS authors have the option to publish the peer review history of their article (what does this mean?). If published, this will include your full peer review and any attached files.

Reviewer #2: No

---

## [Editor Report · Acceptance letter]

26 Aug 2024

PONE-D-24-06837R1 

PLOS ONE

Dear Dr. Ye, 

I'm pleased to inform you that your manuscript has been deemed suitable for publication in PLOS ONE. Congratulations! Your manuscript is now being handed over to our production team.

Kind regards, 

on behalf of

Dr. Wajid Khan 

Academic Editor

PLOS ONE